# Oxidative post-translational modification of EXECUTER1 is required for singlet oxygen sensing in plastids

Vivek Dogra [1,3], Mingyue Li [1,2,3], Somesh Singh[1,3], Mengping Li[1,2] & Chanhong Kim [1,2]

Environmental information perceived by chloroplasts can be translated into retrograde signals that alter the expression of nuclear genes. Singlet oxygen ($^1O_2$) generated by photosystem II (PSII) can cause photo-oxidative damage of PSII but has also been implicated in retrograde signaling. We previously reported that a nuclear-encoded chloroplast FtsH2 metalloprotease coordinates $^1O_2$-triggered retrograde signaling by promoting the degradation of the EXE-CUTER1 (EX1) protein, a putative $^1O_2$ sensor. Here, we show that a $^1O_2$-mediated oxidative post-translational modification of EX1 is essential for initiating $^1O_2$-derived signaling. Specifically, the Trp643 residue in DUF3506 domain of EX1 is prone to oxidation by $^1O_2$. Both the substitution of Trp643 with $^1O_2$-insensitive amino acids and the deletion of the DUF3506 domain abolish the EX1-mediated $^1O_2$ signaling. We thus provide mechanistic insight into how EX1 senses $^1O_2$ via Trp643 located in the DUF3506 domain.

[1] Shanghai Center for Plant Stress Biology and Center of Excellence in Molecular Plant Sciences, Chinese Academy of Sciences, 200032 Shanghai, China. [2] University of the Chinese Academy of Sciences, 100049 Beijing, China. [3] These authors contributed equally: Vivek Dogra, Mingyue Li, Somesh Singh. Correspondence and requests for materials should be addressed to C.K. (email: chanhongkim@sibs.ac.cn)

Singlet oxygen ($^1O_2$) is one of the chloroplast-produced reactive oxygen species (ROS). Excess light drives $^1O_2$ generation in both the light-harvesting antenna complex and in photosystem II (PSII) reaction center (RC) in which excited chlorophyll (i.e., triplet state of chlorophyll) molecules provide light energy to produce $^1O_2$[1,2]. $^1O_2$ has long been considered as a toxic oxygen molecule because of its extremely short lifespan and high reactivity[1]. This mainly PSII-derived highly reactive $^1O_2$ oxidizes macromolecules, especially PSII proteins and associated β-carotene, as well as unsaturated fatty acids[3–7]. PSII core protein D1 is known as a prime target of $^1O_2$ owing to its proximity to the site of $^1O_2$ generation. An increasing body of evidence, however, clearly indicates a signaling role of $^1O_2$, which primes a wide spectrum of biological processes such as acclimation, growth inhibition, and cell death. Two fascinating *Arabidopsis* mutants, *chlorina 1* (*ch1*) and *fluorescent* (*flu*), provided significant insights into $^1O_2$-signaling pathways. *Ch1* encodes the chlorophyll a oxidase (CAO) and its inactivation leads to Chl *b* deficiency, resulting in the loss of the PSII antenna complex[8,9]. This causes an enhanced photoinhibition in PSII RC under light stress. In contrast to *ch1*, for which excess light is required to generate $^1O_2$, in the PSII RC, *flu* is a conditional mutant generating $^1O_2$ upon a dark-to-light shift[10,11]. While $^1O_2$ is generated under this special condition, no other ROS such as hydrogen peroxide ($H_2O_2$)[12] is produced that antagonizes $^1O_2$-triggered signaling[13]. Hence, dissecting $^1O_2$-triggered retrograde signaling can be achieved under both photoinhibitory and non-photoinhibitory conditions using *ch1* and *flu*, respectively.

Detailed studies on these mutants have established not only the signaling role of $^1O_2$ but also two putative $^1O_2$ sensors, namely β-carotene and the EXECUTER1 (EX1) protein[8,14–17]. Recent studies on these $^1O_2$ sensors uncovered their spatial separation in thylakoids[18–20]. While β-carotene molecules function in the grana core (appressed regions of grana) where active PSII dimers produce $^1O_2$, most of EX1 proteins accumulate in the grana margin (non-appressed regions of grana)[18]. EX1 proteins seem to be associated with the PSII repair machinery, including FtsH proteases, chlorophyll synthesis enzymes, and proteins involved in de novo protein synthesis. Interestingly, FtsH2 protease appears to coordinate $^1O_2$ signaling by promoting EX1 degradation upon release of $^1O_2$[18–20]. This finding raised an intriguing hypothesis that EX1 may sense $^1O_2$ released during the PSII repair in the grana margin[19,21]. This also leads us to reconsider an archetypal view that the PSII complex in the grana core is a major source of $^1O_2$ generation. Perhaps, the grana margin-generated $^1O_2$ and EX1 also play an important role under non-photoinhibitory stress conditions such as pathogen attack. Recent studies showed an ROS burst in chloroplasts after the perception of a bacterial pathogen[22–24]. Such stress may not activate the β-carotene-mediated retrograde signaling pathway, which occurs under excess light conditions, but rather activates EX1-mediated $^1O_2$ signaling. Exploration of this assumption may provide us a clue about why chloroplasts contain two distinct and spatially separated $^1O_2$ sensors.

Excess light drives the oxidation of β-carotene, leading to the generation of the volatile retrograde signaling molecule β-cyclocitral (β-CC)[4]. A recent report showed that β-CC induces the expression of genes encoding chloroplast-localized detoxification proteins via TGAII/Scarecrow-like protein 14 (SCL14) transcription factors[25]. In contrast, although EX1 was revealed as a putative sensor mediating $^1O_2$ signaling in both *flu* and wild-type (WT) plants[26,27], its mode of action is largely unknown. It is noteworthy that proteins are prime targets of $^1O_2$ besides other biomolecules, including nucleic acids, lipids, quinones, and isoprenoids[28–31]. Therefore, it is tempting to assume that EX1 protein undergoes $^1O_2$-dependent oxidative modification, which

might be essential for its subsequent proteolysis and signal transduction. In accordance with this notion, we verified in this study that EX1 protein undergoes oxidative modification in a $^1O_2$-dependent manner and that the oxidation at Trp643 is pivotal to initiate the EX1-mediated $^1O_2$-signaling pathway.

## Results

**Light-dependent EX1 degradation.** Previously, we demonstrated that EX1 proteins undergo a rapid degradation upon the burst of $^1O_2$, suggesting a probable modification of EX1 in response to $^1O_2$, priming its subsequent degradation. This assumption also suggests a gradual accumulation of EX1 in the dark because of lack of $^1O_2$ in chloroplasts. To explore this assumption, we analyzed the steady-state levels of EX1 proteins using 5-day-old *ex1* transgenic seedlings expressing GFP-tagged EX1 driven by the CaMV 35S (35S) promoter subjected to varying lengths of darkness followed by illumination (Fig. 1a). Accordingly, we found that EX1 proteins markedly accumulated in the dark, proportionally with the length of darkness, while a significant portion of EX1 proteins was degraded upon re-illumination (Fig. 1a, c). This light-induced proteolysis of EX1 was further reinforced in the *flu* mutant background (*EX1-GFP ex1 flu*) (Fig. 1b, c), suggesting that $^1O_2$ may stimulate EX1 proteolysis through post translational modification (PTM). As anticipated, FtsH2 protease was required to promote EX1 degradation upon illumination (Supplementary Fig. 1a–c), linking the EX1 degradation with $^1O_2$. $^1O_2$ is a byproduct of photosynthesis and the PSII repair is a default process required in the presence of light regardless of light intensity[32]. Therefore, it is not surprising to observe the degradation of EX1 upon exposure to light in the WT background (i.e., *EX1-GFP ex1*). The absence of stress symptoms in WT following a dark-to-light shift with the gradual EX1 degradation suggests that there is a certain threshold level of EX1 turnover or $^1O_2$ for initiating $^1O_2$ signaling. Nonetheless, given that the membrane-bound FtsH protease, whose function is mainly implicated in PSII repair, coordinates $^1O_2$ signaling in *flu* by promoting the degradation of EX1 upon release of $^1O_2$[18–20] and the enhanced levels of $^1O_2$-promoted EX1 degradation in *flu* (Fig. 1b, c), all these findings inspired us to examine any PTM in EX1 associated with $^1O_2$ generation.

**Oxidative PTM of EX1 in a light-/$^1O_2$-dependent manner.** In an attempt to reveal possible PTM(s) in EX1 upon release of $^1O_2$, the *EX1-GFP ex1 flu* transgenic seedlings grown under dark or continuous light condition (CL hereafter) were used to enrich EX1 protein using a GFP-Trap coupled to magnetic agarose beads. We anticipated that EX1 extracted from the dark-grown seedlings would avoid $^1O_2$-dependent PTM since the generation of $^1O_2$ in chloroplasts relies on light. The significantly enriched EX1-GFP proteins were established from both dark- and CL-grown seedlings (Fig. 2a and Supplementary Data 1). The PSII core proteins, light-dependent NADPH:protochlorophyllide oxidoreductase (POR), and FtsH2 protease co-immunoprecipitated with EX1-GFP in the chloroplasts, coinciding with a previous report[15], while POR (most likely PORA based on the MS data; Supplementary Data 1) and FtsH2 protease co-immunoprecipitated with EX1-GFP in the etioplasts (Fig. 2a). The enriched EX1-GFP and associated proteins were subsequently subjected to a mass-spectrometric (MS) analysis to examine EX1 modification. Tandem MS analysis revealed a significant coverage for EX1 protein in both dark (~40%) and CL (~53%) samples (Supplementary Fig. 2a and Supplementary Data 1). We scanned the MS results for any global PTMs, including phosphorylation, acetylation, and oxidations using the Mascot search engine. As a result, we found one specific

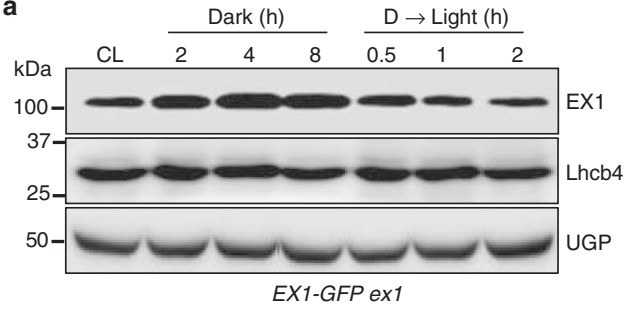

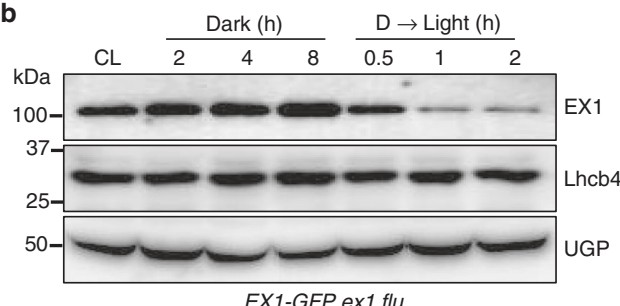

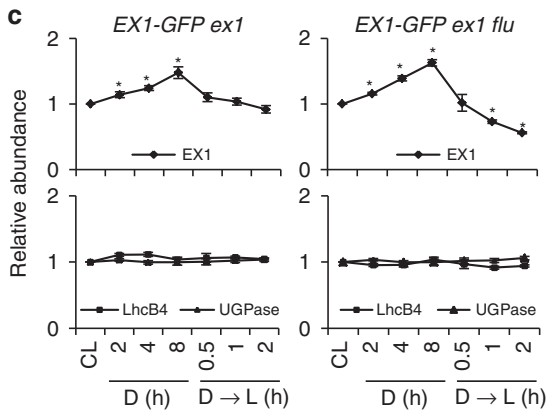

**Fig. 1** Light- and $^1O_2$-dependent EX1 degradation. Continuous light (CL)-grown 5-day-(d)-old transgenic seedlings of **a** *ex1* and **b** *ex1 flu* expressing EX1-GFP under the control of the 35S promoter were transferred to the dark (for 2, 4, 8 h) and the 8-h-dark (D)-treated seedlings were re-exposed to light (for 0.5, 1, 2 h) at the light intensity of 100 μmolm$^{-2}$s$^{-1}$. Total protein was extracted and analyzed by western blot. Chlorophyll a/b binding protein CP29 (Lhcb4) and cytosolic UDP-glucose pyrophosphorylase (UGP) were used as controls. EX1-GFP, Lhcb4, and UGPase were detected using antibodies against GFP, Lhcb4, and UGP, respectively. **c** The levels of EX1-GFP in the dark or after re-exposing to light were compared to its abundances under CL conditions. Average intensity values of the protein bands were calculated using AzureSpot software v14.0 (AZURE). Data represents the mean of three biological repeats. Error bars show standard error of the mean. Asterisks in **c** indicate statistically significant differences to CL condition ($P < 0.05$, Student's *t*-test)

modification, namely oxidative PTM (Oxi-PTM hereafter) at a Trp (W) residue in peptide $^{643}$W$^{ox}$VDGELVILDGK$^{654}$ of EX1 in the CL-grown seedlings (Fig. 2b). By contrast, no oxidation of Trp643 was observed in EX1 isolated from dark-grown seedlings indicating that it occurred in a light-dependent manner. Previous studies demonstrated that PSII core proteins such as D1, D2, and CP43 proteins undergo Oxi-PTM specifically at certain Trp residues under high-light stress[33–35], which was therefore suggested as a plausible signature of photoinhibition. Oxidation of Trp results in the modification of its indole side chain into the

keto-amino-hydroxy derivative oxindolylalanine (Oia), a dihydro-hydroxy derivative *N*-formylkynurenine (NFK), and kynurenine with the corresponding mass shifts of $+16$, $+32$, and $+4$ Da, respectively (Fig. 2b)[35]. We confirmed all these Trp643 variants in EX1 isolated from the CL-grown seedlings (Fig. 2b and Supplementary Table 1). Not only EX1, but also the PSII core proteins that interact with EX1 in the grana margin, contained these Trp oxidations (Table 1) consistent with earlier reports[33]. It is noteworthy that we also found oxidized Trp residues in D1, D2, and CP47, as shown in Table 1. Interestingly, most of the Trp residues undergoing oxidation in EX1 and the PSII core proteins are surrounded or followed by smaller amino acids such as glycine, alanine, valine, leucine (carrying nonpolar aliphatic side chains), and serine. This might be crucial for the accessibility of the Trp residue for oxidation. Given that our transgenic plants were grown under moderate light intensity (100 μmol m$^{-2}$ s$^{-1}$), it is very likely that the Trp oxidation of PSII core proteins as well as of EX1 occurs regardless of the light intensity above a certain threshold. This is also consistent with the notion that $^1O_2$ is a byproduct of photosynthesis and that PSII repair is a default process. The notable increase in the ratio of oxidized Trp643 to the non-oxidized one in the EX1 pool enriched from the *flu* mutant seedlings subjected to a dark-to-light shift further suggests that Trp643 might undergo Oxi-PTM by $^1O_2$ (Fig. 2c).

Among the $^1O_2$-sensitive amino acid residues carrying double bonds or reactive thiol groups in their side chains, only Trp can scavenge $^1O_2$ via both physical quenching ($k = 2–7 \times 10^7$ L mol$^{-1}$ s$^{-1}$), as well as chemical reactions ($k = 3.2 \times 10^7$ L mol$^{-1}$ s$^{-1}$ at physiological pH)[29,30]. Thus, Trp is highly susceptible and reacts rapidly with $^1O_2$[29,30]. While Trp is relatively rare in proteins, it is quite abundant in membrane proteins as well as in peripheral membrane proteins where it plays an anchoring role in the lipid-water interface[36]. In addition, Trp forms clusters with side chains of other residues, which is important for protein folding, structural stability, and protein interactions[36]. As the oxidative modifications of Trp disrupt the heterocyclic indole ring of the side chain (Fig. 2b), the aforementioned typical roles of Trp could be interrupted. Indeed, as demonstrated previously, $^1O_2$-induced oxidations alter the physical and chemical properties (e.g., hydrophobicity) of proteins, which may lead to their aggregation or fragmentation[29,30]. In humans, these events have been implicated with cellular and tissue dysfunction, including apoptosis, necrosis, and altered cell signaling[29,30,37,38]. Besides, proteins undergoing tryptophan oxidation have also been found in ROS-producing mitochondria, involved in mediating retrograde signaling[39,40].

**Trp643 is located in domain of unknown function 3506.** The EX1 protein contains two predicted domains namely UVR (UvrB/C; http://www.ebi.ac.uk/interpro/entry/IPR001943) near the N-terminus of the mature form of EX1 and the domain of unknown function 3506 (DUF3506; http://www.ebi.ac.uk/interpro/entry/IPR021894) at the C-terminal end (Fig. 3a). To date, the biological functions of these two domains have not been elucidated. EX1 contains 11 Trp residues at positions 67, 88, 111, 113, 116, 198, 199, 264, 565, 643, and 667 (Supplementary Fig. 2b), of which Trp565 and Trp643 are located within the DUF3506 domain (Fig. 3a, b). In order to examine whether both Trp residues in the DUF3506 domain are conserved, the sequences of the EX1-like protein from 27 plant species reported in Phytozome v12.1 were aligned using ClustalW. Notably, the resulting alignment verified that both Trp565 and Trp643 are absolutely conserved in plants (Fig. 3b). In addition, the DUF3506 domain is also highly conserved across different plant species (Fig. 3b). Interestingly, Trp565 remained unchanged

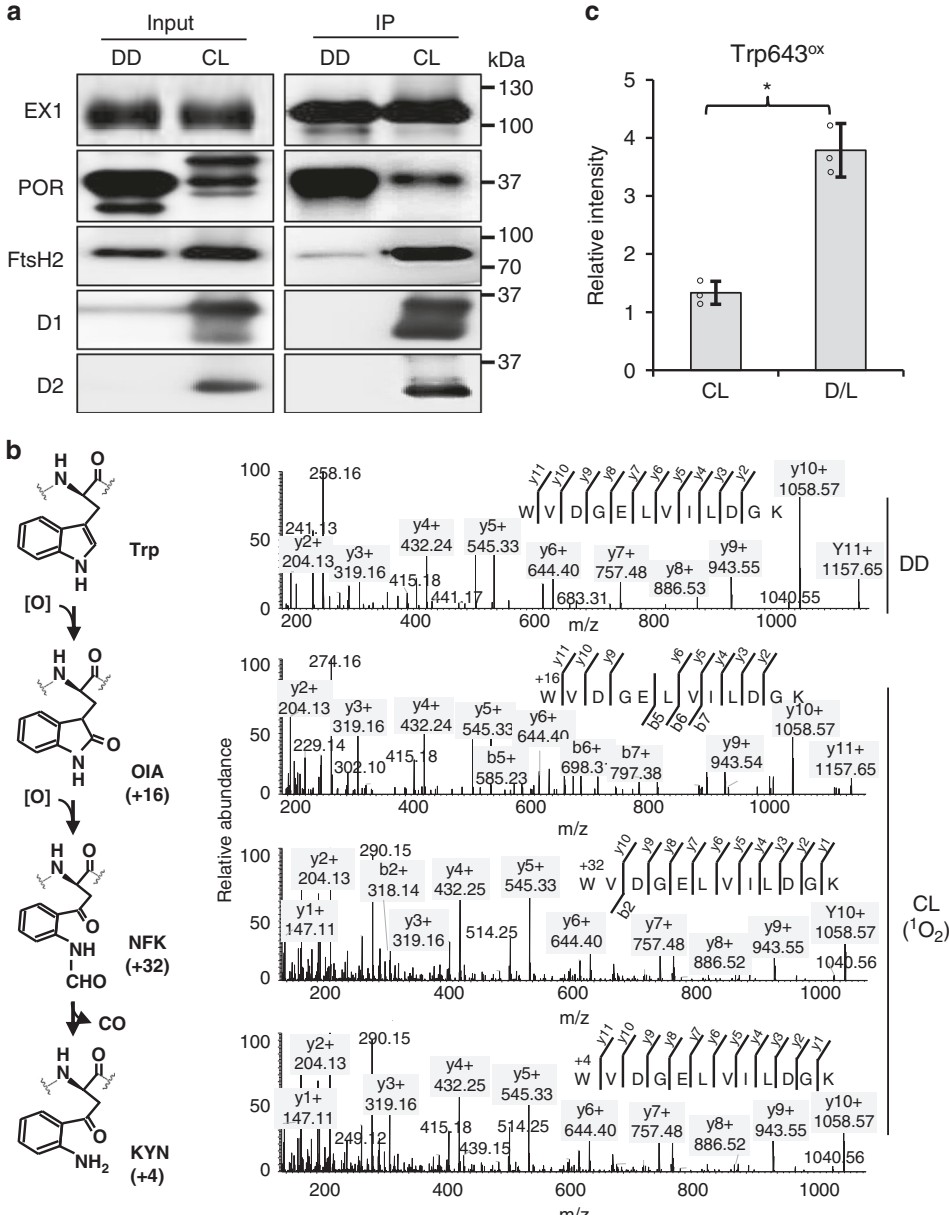

**Fig. 2** Light- and $^1O_2$-dependent Trp643 oxidation occurs in vivo. **a** EX1-GFP protein enrichment. For input, total proteins extracted from 5-day-old dark (DD)-grown or CL-grown seedlings of *EX1-GFP ex1* were analyzed. For IP, immune-reactive EX1-GFP proteins were precipitated from each input sample by adding magnetic agarose beads conjugated with GFP antibody. Trapped proteins were solubilized and separated by SDS-PAGE and probed on western blots using the indicated antibodies. **b** Oxidative PTM (Oxi-PTM) analysis. The IP samples were subjected to PTM analysis by mass spectrometry (MS). EX1 was found to undergo light- and $^1O_2$-dependent oxidation, especially at Trp643 in peptide [643]WVDGELVILDGK[654]. This oxidation led to the formation of oxindolylalanine (OIA), *N*-formylkyrnurenine (NFK), and kynurenine (KYN) with $+16$, $+32$, and $+4$ mass shifts, respectively. **c** The effect of increasing levels of $^1O_2$ on Trp643 oxidation. To increase the levels of $^1O_2$, 5-day-old seedlings of *EX1-GFP ex1 flu* grown under CL were subjected to 2 h dark and then re-exposed to light for 5 min (D/L). The GFP-trapped EX1-GFP proteins were subjected to MS analysis and the relative levels of Trp643 oxidation was measured. Oxidation levels were calculated using the sum of absolute intensities of the peptides carrying oxidized Trp643. Bar chart represents the relative intensity of the oxidized peptides. Asterisks indicate statistically significant differences between the mean values ($P < 0.05$, Student's *t*-test)

(Fig. 3c) while only Trp643 was oxidized, which underlines the specificity of Trp643 in the context of Oxi-PTM. Considering that Trp643 is located in the highly conserved DUF3506 domain and that Trp643 is likely to interact with $^1O_2$, leading to its oxidation, DUF3506 possibly has a role in sensing $^1O_2$. Therefore, here we renamed DUF3506 as a singlet oxygen sensor (SOS) domain.

**Trp643 is required for both EX1 stability and $^1O_2$ signaling.** To test whether Trp643 oxidation is an integral step in activating $^1O_2$

signaling, we substituted Trp643 with leucine (Leu, L) or alanine (Ala, A) that are both known to be $^1O_2$-insensitive amino acids and were previously used to substitute the Oxi-PTM Trp residue of the CP43 protein, a constituent of the PSII RC[35]. Anderson et al.[35] explored the biological relevance of this oxidation by substituting Trp352 with Leu, Ala, or Cys (cysteine, C)[35]. Each substitution led to an increased photoinhibition as was evident from an increased photodamage and decreased turnover of PSII. As Cys is a potential target of $^1O_2$[30], in this study we opted for

**Table 1 Trp oxidation in EX1 and associated PSII proteins**

| Protein | Sequence | AA modified | Observed mass (m/z) | Actual mass (Da) | Charge (z) | Error (ppm) | Prob. (%) |
|---|---|---|---|---|---|---|---|
| EX1 | $^{643}$RW$^{ox}$VDGELVILDGK$^{654}$ | Trp643 | 688.36 | 1374.71 | 2+ | 4.7 | 100 |
| D1 | $^{9}$ESESLW$^{ox}$GR$^{16}$ | Trp14 | 498.23 | 994.44 | 2+ | 2.6 | 99 |
| | $^{313}$VINTW$^{ox}$ADIINR$^{323}$ | Trp317 | 673.86 | 1345.70 | 2+ | 2.9 | 100 |
| D2 | $^{14}$DLFDIMDDW$^{ox}$LR$^{24}$ | Trp22 | 735.84 | 1469.66 | 2+ | 7.3 | 100 |
| | $^{328}$AW$^{ox}$MAAQDQPHENLIFPEEVLPRG$^{349}$ | Trp329 | 875.09 | 2622.26 | 3+ | 2.3 | 100 |
| CP43 | $^{363}$APW$^{ox}$LEPLR $^{370}$ | Trp365 | 507.28 | 1012.54 | 2+ | 7.1 | 96 |
| | $^{383}$DIQPW$^{ox}$QER$^{390}$ | Trp387 | 552.26 | 1102.51 | 2+ | 2.1 | 96 |
| CP47 | $^{273}$YQW$^{ox}$DQGYFQQEIYR$^{286}$ | Trp275 | 978.44 | 1954.86 | 2+ | 6.5 | 100 |
| | $^{288}$VSAGLAENQSLSEAW$^{ox}$AK$^{304}$ | Trp302 | 896.95 | 2687.83 | 3+ | 180 | 99 |

MS-based PTM analysis identified all three Trp oxidation variants, but the parameters for the double oxidation-induced NFK modifications with a mass shift of + 32, have been shown here

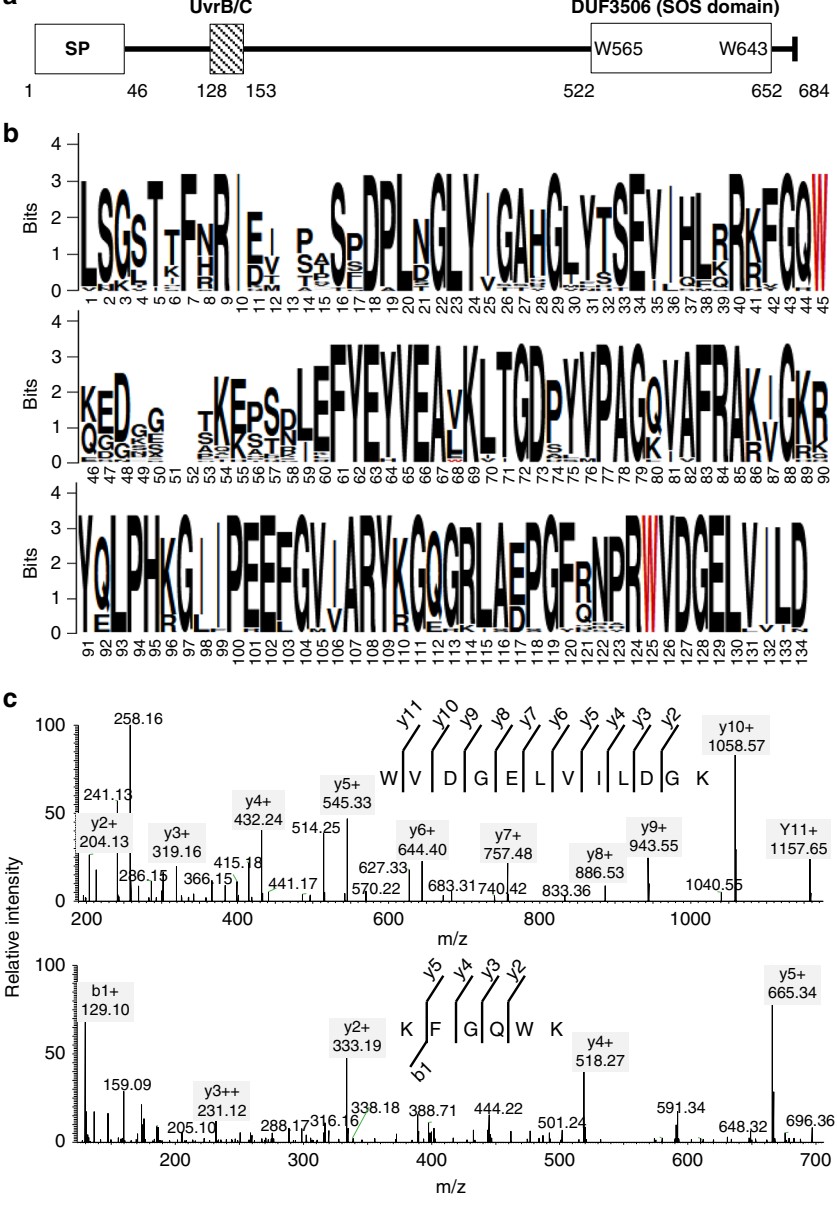

**Fig. 3** Trp643 is located in the DUF3506 domain. **a** EX1 protein comprises two predicted domains, UvrB/C and DUF3506. **b** Multiple alignment reveals that the DUF3506 domain is highly conserved across different plant species. The sequences of the EX1-like protein from the 27 plant species reported in Phytozome v12.1 were aligned using ClustalW. Two Trp residues, Trp565 and Trp643, were highlighted in red. Aligned DUF3506 domain sequences were visualized using WEBLOGO. **c** Tandem MS spectra of peptides $^{562}$GQWKGGK$^{569}$ and $^{643}$W$^{ox}$VDGELVILDGK$^{654}$ revealed the specificity of Trp643 in the context of Oxi-PTM. This experiment was repeated using three independent biological samples with reproducible results

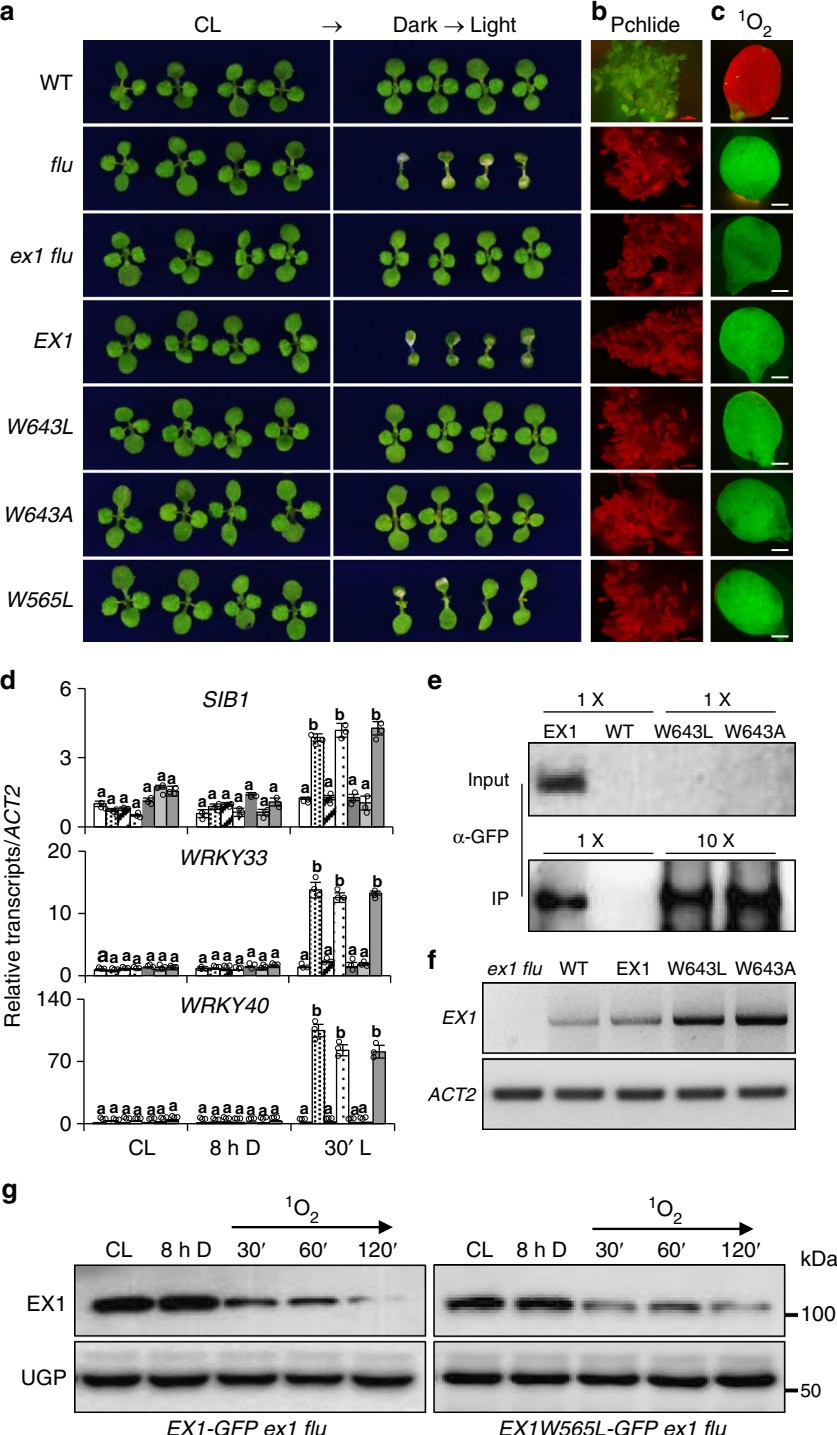

Leu and Ala to substitute Trp643 in EX1. The GFP-tagged modified versions of EX1, EX1W643L, and EX1W643A were expressed in *ex1 flu* under the control of the 35S promoter to examine whether they are able to complement *ex1 flu* in regaining the typical *flu* mutant phenotype. In contrast to EX1, both modified EX1s failed to complement *ex1 flu* (Fig. 4a–d). Despite the comparable levels of Pchlide in the dark and $^1O_2$ upon re-illumination relative to *flu* (Fig. 4b, c), neither EX1W643L nor EX1W643A was able to restore the $^1O_2$-triggered cell death[10,15] and the expression of $^1O_2$-responsive genes (SORGs)[20]. By contrast, the wild-type EX1 (EX1-GFP) expressed in *ex1 flu* entirely restored the *flu*-conferred phenotypes (Fig. 4a, d). Unexpectedly,

we found that these modified proteins are highly unstable as revealed by the western blot analysis, contrasting with the higher levels of their transcripts as compared to *EX1-GFP* (Fig. 4e, f). We could not detect any fusion proteins by confocal microscopy and by western blot analysis in over 48 individual transgenic lines (data not shown). Even though the proteins were enriched using chloroform-methanol and a large amount of proteins (720 μg) were loaded for each sample, modified EX1-GFP proteins were still undetectable. In an effort to detect these fusion proteins, proteins were enriched with the GFP-Trap, using 5 and 25 mg of total proteins to concentrate EX1-GFP and modified EX1-GFP, respectively. The trapped proteins were eluted and denatured in

**Fig. 4** Substitution of Trp643 with $^1O_2$-insensitive amino acids inactivates $^1O_2$ signaling. **a** Five-day-old seedlings of wild-type (WT), *flu*, *ex1 flu*, *EX1-GFP ex1 flu* (EX1), *EX1W643L-GFP ex1 flu* (W643L), *EX1W643L-GFP ex1 flu* (W643A), and *EX1W565L-GFP ex1 flu* (W565L) grown under CL were transferred to long-day condition (LD; 16:8-h light–dark cycle) for 3 days. Seedlings grown under CL for 8 days were shown as controls. **b** Pchlide accumulation. Five-day-old dark-grown seedlings were exposed to blue light (400–450 nm) and the emitted red fluorescence was recorded under the fluorescent microscope. **c** Singlet oxygen sensor green (SOSG) fluorescence in 5-day-old CL grown seedlings subjected to 8 h dark followed by 30 min re-illumination. Fluorescence of $^1O_2$-activated SOSG was recorded using GFP filter. The red fluorescence derives from the chlorophyll in the chloroplasts. **d** The relative expression levels of $^1O_2$-responsive genes (SORGs) including *SIGMA FACTOR BINDING PROTEIN 1* (*SIB1*), *WRKY40*, and *WRKY33* were analyzed in 5-day-old seedlings grown under CL, subjected to 8 h dark (8 h D) and then followed by re-illumination for 30 min (30′ L) using qRT-PCR. *ACT2* was used as an internal control. Data represents the mean of three independent biological replicates. Error bars indicate standard deviation. Lower case letters indicate statistically significant differences between the mean values ($P < 0.05$, one-way analysis of variance with post hoc Tukey's Honest Significant Difference test). **e, f** The protein abundance of EX1W643A/L was uncoupled from its transcript levels. **e** To detect EX1W643A/L-GFP proteins by western blot, up to 720 μg of chloroform-methanol-enriched proteins were loaded. For CoIP/western blot, 5 and 25 mg of total proteins were used to detect EX1-GFP and EX1W643A/L-GFP proteins, respectively, and proteins were eluted in 100 μL of loading buffer. Twenty microliters of IP sample were loaded in case of EX1-GFP whereas 80 μL were loaded for EX1W643A/L-GFP. **f** Total RNA was isolated from 5-day-old seedlings grown under CL and the transcript levels of EX1 (or EX1W643L/A-GFP) were detected by semi-quantitative RT-PCR. *ACT2* was used as an internal control. **g** EX1W565L-GFP proteins were rapidly degraded upon a dark-to-light shift similar to EX1-GFP. UGP was used as a loading control

100 μL of loading buffer. Afterward, 20 μL (EX1-GFP) and 80 μL (modified EX1-GFP) of eluted proteins were used to detect the corresponding proteins by western blot analysis. As a result, we were able to detect both EX1W643A- and EX1W643L-GFP proteins along with EX1-GFP (Fig. 4e), indicating that modified EX1 proteins were translated but failed to persist. Next, to ensure their targeting into the chloroplasts, the Trp643-substituted constructs were expressed in *Nicotiana benthamiana* leaves. The GFP signals of EX1W643L- and EX1W643A-GFP were clearly detected exclusively in the chloroplasts (Supplementary Fig. 3), indicating that there were no negative effects of the substitutions on their subcellular localization. This result also indicates that the modified EX1 proteins, though stably expressed in Arabidopsis, are extremely unstable in contrast to those expressed in mature *N. benthamiana* leaves.

Considering that EX1 degradation is essential for initiating $^1O_2$ signaling[18,20], it is puzzling that despite being constantly degraded, the Trp643-substituted EX1 proteins failed to mediate $^1O_2$ signaling. The uncoupling of $^1O_2$ signaling from the EX1 degradation would be explained in that the Trp643 oxidation and EX1 degradation are coordinated with a release of a yet unknown retrograde signaling molecule, possibly via an EX1-associated protein complex in the grana margin. Perhaps, the Trp643-substituted EX1 proteins (EX1W643L and EX1W643A) may not achieve this condition to initiate $^1O_2$ signaling. Alternatively, the initiation of EX1-mediated $^1O_2$ signaling may require a specific signalosome, including FtsH protease. Therefore, it is possible that both EX1W643L and EX1W643A proteins may undergo rapid proteolysis via a different signalosome, which may consequently impair $^1O_2$ signaling.

The second Trp residue in the SOS domain, Trp565, was also substituted with Leu to examine the specificity of Trp643 toward $^1O_2$ signaling. EX1W565L-GFP completely complemented *ex1 flu* as evidenced by the emergence of cell death and upregulation of SORGs upon a dark-to-light shift (Fig. 4a, d). Afterward, we examined the stability of EX1W565L-GFP upon a dark-to-light shift in comparison with EX1-GFP (Fig. 4g and Supplementary Fig. 4a). The western blot result found a similar stability of EX1W565L-GFP proteins as for EX1 (Fig. 4g and Supplementary Fig. 4a), indicating no link between Trp565 and $^1O_2$.

**The SOS domain controls EX1 stability and $^1O_2$ signaling.** Given that $^1O_2$ promoted Trp643 oxidation in the SOS domain and that the modified EX1 proteins carrying a Trp643 substitution failed to accumulate, examining a truncated EX1 lacking the SOS domain (EX1ΔSOS) could alternatively

provide insights into the biological importance of the Trp643 oxidation and the SOS domain with respect to $^1O_2$ signaling. If the Trp643 oxidation primes the subsequent EX1 degradation, the deletion of the SOS domain would abolish both the degradation of EX1ΔSOS and the $^1O_2$-triggered retrograde signaling in the *ex1 flu* mutant upon release of $^1O_2$. To explore this hypothesis, we first generated *ex1 flu* stable transgenic lines expressing EX1ΔSOS-GFP under the control of the 35S promoter and examined the stability of EX1ΔSOS-GFP proteins, as well as the relative expression levels of SORGs in response to $^1O_2$. While EX1-GFP complemented *ex1 flu*, EX1ΔSOS-GFP failed to mediate $^1O_2$-triggered stress responses including cell death despite the fact that the transgenic plants accumulated comparable levels of Pchlide in the dark and $^1O_2$ upon re-illumination (Fig. 5a–c). Consistent with the $^1O_2$-insensitive phenotype and the above hypothesis, EX1ΔSOS-GFP proteins remained unchanged upon release of $^1O_2$ and the expression of SORGs was impaired (Fig. 5d, e). Collectively, these findings reassure the proposition that DUF3506 functions as an SOS domain.

An earlier forward genetic study also highlighted the importance of the SOS domain in the context of $^1O_2$ signaling[15]. Three missense mutations in EX1 repressed the $^1O_2$ signaling. Among them, two were located in the SOS domain resulting in the substitution of Phe528Cys and Gly646Asp, respectively[15]. To test whether the impaired $^1O_2$ signaling in *EX1F528C flu* and *EX1G643D flu*[15] resulted from the uncoupled Trp643 oxidation to $^1O_2$, we first generated stable transgenic lines expressing EX1F528C-GFP and EX1G646D-GFP under the control of the 35S promoter in the *ex1 flu* background. In contrast to EX1-GFP, both EX1F528C-GFP and EX1G646D-GFP failed to complement *ex1 flu* (Fig. 5a). Like EX1ΔSOS-GFP, EX1F528C-GFP and EX1G646D-GFP proteins remained unchanged upon release of $^1O_2$ in the chloroplasts (Fig. 5e and Supplementary Fig. 4b) and the $^1O_2$ signaling was compromised as evidenced by the impaired responses of the SORGs (Fig. 5a–d). The subsequent MS analysis identified the Trp643 oxidation in EX1-GFP proteins, however, no Trp643 oxidation was detected in EX1F528C-GFP and EX1G646D-GFP proteins (Supplementary Figs. 5 and 6 and Supplementary Table 2). This result supports the notion that Trp643 is required to sense $^1O_2$ and to initiate $^1O_2$ signaling.

**The SOS domain is associated with the signalosome.** The thylakoid membrane-anchored EX1 is likely to sense $^1O_2$, involving Trp643 oxidation, followed by a rapid proteolysis. Since Trp also mediates hydrophobic interactions[30], it is possible that Trp643 plays an important role in sensing $^1O_2$ and/or attaching the SOS domain to the thylakoid membranes. Alternatively, the SOS

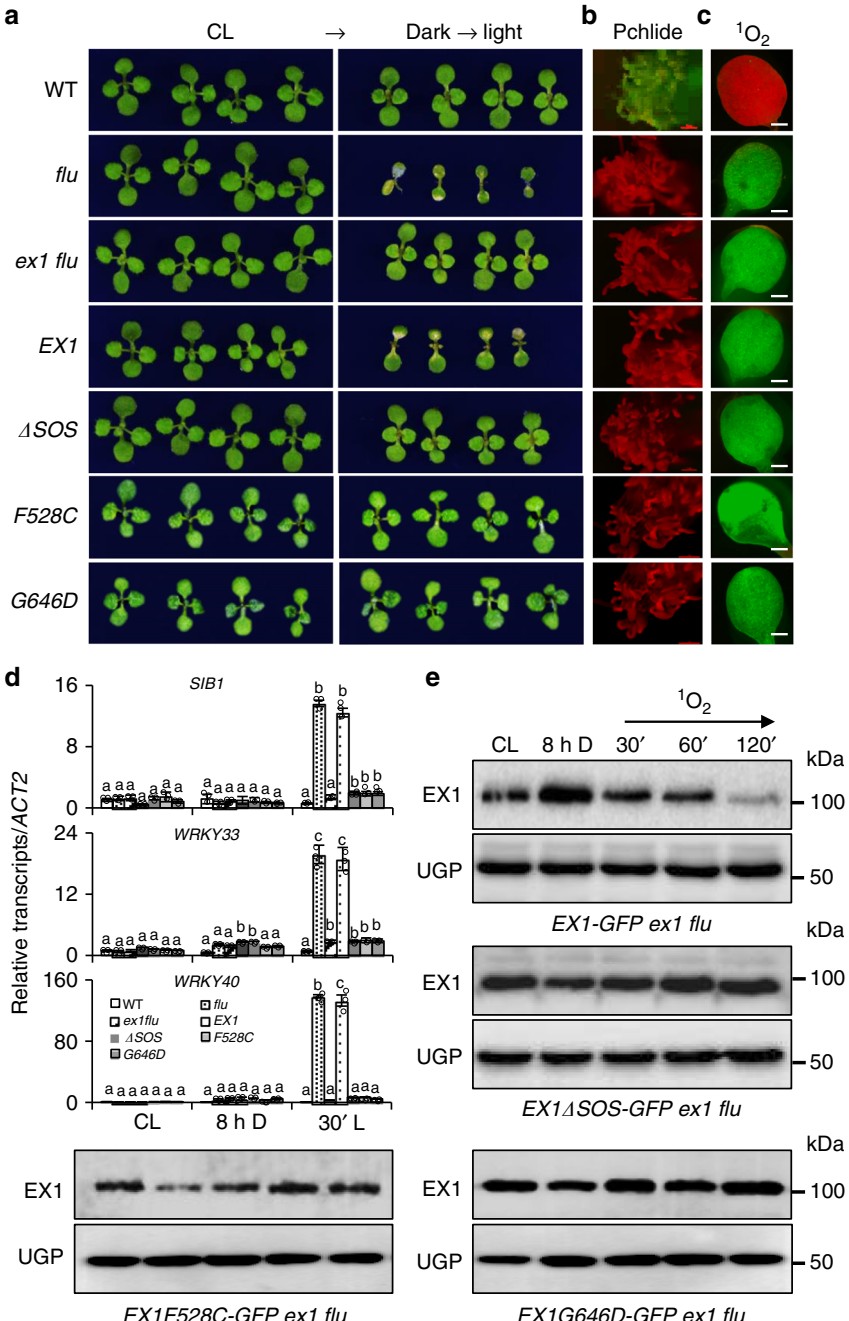

**Fig. 5** The SOS domain is essential for EX1 degradation and $^1O_2$ signaling. **a** EX1ΔSOS-GFP, EX1F528C-GFP, and EX1G646D-GFP are unable to complement *ex1 flu*. Seedlings of WT, *flu*, *ex1 flu*, EX1-GFP *ex1 flu* (*EX1*), EX1ΔSOS-GFP *ex1 flu* (*ΔSOS*), EX1F528C-GFP *ex1 flu* (*F528C*), and EX1G646D-GFP *ex1 flu* (*G646D*) grown for 5 days under CL were shifted to LD conditions for 3 days. Seedlings grown under CL for 8 days serve as controls. **b** Pchlide accumulation in etiolated seedlings. **c** Singlet oxygen sensor green (SOSG) fluorescence and **d** expression of SORGs were determined as shown in Fig. 3c, d. *ACT2* was used as an internal control. Data represents the mean of three independent biological replicates. Error bars indicate standard deviation. Lower case letters indicate statistically significant differences between the mean values (*P* < 0.05, one-way analysis of variance with post hoc Tukey's Honest Significant Difference test). **e** Deletion or modification of the SOS domain prevents the EX1 degradation. Total protein isolated from 5-day-old seedlings initially grown under CL and then subjected to 8 h dark followed by 30-, 60-, and 120-min re-illumination, respectively, were analyzed by western blot using GFP antibody. UGP was used as a loading control

domain may attach EX1 to the membrane via an interaction with membrane proteins such as PSII core proteins (EX1 was found to interact with several PSII core proteins; see Wang et al.)[18], which may expose the side chain of Trp643 to $^1O_2$. In either case, without the SOS domain, EX1ΔSOS-GFP should accumulate in the stroma. Consistently, the fractionation of chloroplasts isolated from EX1ΔSOS-GFP and EX1-GFP transgenic plants revealed

that most of the EX1ΔSOS-GFP proteins were localized in the stroma, while EX1-GFP proteins resided in the membrane fraction (Supplementary Fig. 7). Considering the remarkable stability of EX1ΔSOS-GFP proteins even in the presence of $^1O_2$ and the accumulation of EX1-GFP in the dark despite the close proximity to the FtsH protease[18], it is reasonable to propose that the FtsH protease or another signalosome involved in EX1 degradation

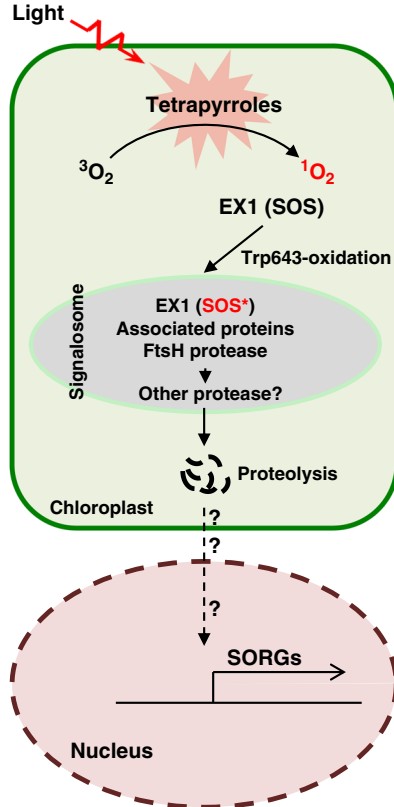

**Fig. 6** EX1-mediated $^1O_2$ signaling. Free tetrapyrrole molecules such as chlorophyll and its precursors are potential sources of $^1O_2$ generation. In the *flu* mutant, $^1O_2$ oxidizes the Trp643 residue in the SOS domain of EX1. The oxidized EX1 protein, perhaps also EX1-associated proteins, is recognized by the signalosome where the thylakoid membrane-bound FtsH protease promotes EX1 degradation. This process is a prerequisite for initiating the $^1O_2$ signaling. The degradation of EX1 and possibly its associated proteins may release a yet unknown retrograde signaling molecule, which induces the expression of SORGs

may recognize it as a putative substrate via the oxidized SOS domain for the subsequent proteolysis (Fig. 6). Co-immunoprecipitation of EX1 with FtsH protease and chaperone proteins such as Hsp70 and ClpC1/2 (Supplementary Data 1) is indicative of the existence of such a signalosome.

## Discussion

We previously demonstrated that $^1O_2$-driven EX1 proteolysis via the FtsH protease is an integral part of $^1O_2$-triggered retrograde signaling[18]. The genetic inactivation of FtsH2, a major subunit of the hexameric FtsH protease complex, appeared to abolish the EX1-mediated $^1O_2$-signaling pathway[20]. In this study, we further investigated the underlying molecular mechanism of EX1 degradation, which was considered as an important step towards understanding EX1 function as a putative $^1O_2$ sensor (Figs. 1–5). As a result, we here show a mechanistic insight into how the EX1 protein senses $^1O_2$ with some intriguing proposals, i.e., the SOS (DUF3506) domain and its constituent Trp643 are important for EX1 proteolysis and $^1O_2$ sensing, respectively (Fig. 6).

Recently, we proposed that the grana margin where the PSII repair occurs would be an alternative source of $^1O_2$ generation besides the grana core-assembled PSII[18,19]. This proposal stemmed from the facts that EX1 mainly resides in the grana margin and that $^1O_2$ has an extremely short lifespan (~200 ns), limiting its diffusion distance[19,41]. It is therefore unlikely that $^1O_2$

generated in PSII located in the grana core will reach the grana margin where EX1 mediates $^1O_2$ signaling. Therefore, the revealed EX1 interactome, including PSII core proteins, FtsH protease, chlorophyll synthesis enzymes, and proteins involved in de novo protein synthesis[18], prompted us to suggest that under photo-oxidative stress conditions the grana margin may release $^1O_2$ because of the emergence of free tetrapyrroles including chlorophyll when the PSII repair processes becomes impaired[19]. In fact, $^1O_2$ inactivates the chloroplast elongation factor G involved in the PSII repair at the step where newly synthesized D1 and D2 proteins are reassembled[42,43]. This may consequently impair an entire PSII repair process and release free tetrapyrroles, potent photosensitizers, into the grana margin. In agreement with the notion that the grana margin may release $^1O_2$, in this study we observed that the Trp643 residue in the SOS domain of EX1 undergoes oxidation upon release of $^1O_2$ in chloroplasts and its oxidation seems to be essential for triggering an EX1-mediated $^1O_2$ signaling (Figs. 1–5). The importance of the SOS domain in initiating $^1O_2$ signaling was indirectly shown in the earlier genetic screen aimed to find suppressors for *flu*, whereby two missense mutations in the SOS domain, namely Phe528Cys and Gly646Asp, were shown to impair $^1O_2$ signaling[15]. Because of the proximity of Gly646 to Trp643, we reasoned that the failure of the initiation of $^1O_2$ signaling in *EX1G646D flu* is likely to be linked to the uncoupled Trp643 oxidation by $^1O_2$. Even though Phe528 is distant from Trp643, we thought that it could also impair Trp643 oxidation by affecting the structure of the SOS domain. In agreement with this hypothesis, we observed no Trp643 oxidation in EX1F528C-GFP and EX1G646D-GFP proteins (Supplementary Fig. S6). The Trp residues undergoing oxidation in EX1 and the PSII core proteins are surrounded or followed by smaller amino acids, which seems to be crucial for the accessibility of Trp residue for oxidation. Thus, the substitution of Gly (a nonpolar aliphatic side chain-containing residue) with the polar and charged amino acid Asp is likely to affect the structural positioning/proximity of Trp643 in EX1 toward the source of $^1O_2$. Future elucidation of the EX1 protein structure on the oxidation of Trp643 may shed light on the effect of Gly646Asp and perhaps also Phe528Cys.

Despite the range of results presented here, which evidently shed light on the $^1O_2$-dependent oxidation of the SOS domain and its positive role in $^1O_2$ signaling (Figs. 2, 3, 5), it is still unclear whether the grana margin is an alternative source of $^1O_2$ generation. It is possible that certain hydroperoxides generated upon the oxidation of lipids or other reactive molecules may cause indirect Trp643 oxidation. Surprisingly, the de novo synthesis of chlorophyll during PSII repair under photo-oxidative stress is largely unexplored. Most of the attention has been focused on D1 turnover even though free-chlorophyll molecules act as potent photosensitizer when released from PSII during the repair process. Although recent studies have demonstrated the constant turnover of chlorophyll and pheophytin, along with the turnover of D1 protein[44,45], there is a need for a detailed study to understand how chlorophyll synthesis and channeling are coordinated during PSII turnover under photo-oxidative stress conditions. It was recently found that small light-harvesting-like proteins are able to sequester free tetrapyrroles[46–49], suggesting that they may also have a protective role when chlorophyll synthesis and PSII reassembly become impaired. In addition to free tetrapyrroles, the oxidized forms of Trp, such as NFK and KYN, also serve as photosensitizers generating reactive species[29,30,50]. As the damaged PSII RC core proteins with oxidized Trp residues migrate to the grana margin for repair processes, it is possible that the Trp-derived reactive species can also oxidize the proteins in their proximity, including EX1. All these questions need to be addressed for a better understanding of the

mode of action of EX1 as a putative $^1O_2$ sensor and its nearly exclusive presence in the grana margin.

The chloroplastic $H_2O_2$ sensor 3′-phosphoadenosine 5′-phosphate (PAP) phosphatase SAL1 also functions in retrograde signaling via Oxi-PTM[51]. Under oxidative stress conditions, SAL1 appears to undergo oxidations at Cys119, Cys167, and Cys190, resulting in the formation of intra- (Cys119–190) and inter-molecular (Cys167–190) disulfide bridges[51]. Inactivation of SAL1 results in the accumulation of its substrate PAP, which serves as a retrograde signaling molecule[51,52]. Although a genuine signaling molecule acting in EX1-mediated $^1O_2$ signaling remains unknown, the importance of Oxi-PTM in initiating retrograde signaling is apparent for both EX1 and SAL1. Similarly to the importance of EX1 stability toward $^1O_2$ signaling, the nuclear-encoded chloroplast Genomes Uncoupled 1 (GUN1) protein, which mediates biogenic retrograde signaling pathways and which undergoes continuous turnover via the stromal Clp protease, becomes stable when chloroplasts require GUN1 to coordinate the expression of photosynthesis-associated nuclear genes[53]. Taken together, it appeals that further investigations on Oxi-PTM and/or protein turnover of proteins such as EX1, whose function is primarily implicated in chloroplast-to-nucleus retrograde signaling pathways, will provide mechanistic insights into how these proteins perceive and respond to environmental cues.

## Methods

**Plant materials and growth conditions**. All *Arabidopsis thaliana* seeds used in this study were derived from Columbia-0 (Col-0) ecotype and were harvested from plants grown under continuous light (CL) condition (100 μmol m$^{-2}$ s$^{-1}$ at 22 ± 2 ˚C). Mutants used in this study for *ex1* and *flu* were the same as used in Wang et al.[18]. Seedlings of wild type, *flu*, *ex1*, *ex1 flu*, and transgenic *ex1* and *ex1 flu* seedlings expressing GFP-tagged EX1 (or modified EX1) under the control of the 35S promoter were grown on half strength Murashige and Skoog (MS) medium (Duchefa Biochemie) with 0.8% (w/v) agar. Mature plants were grown on soil for 3 weeks under CL. $^1O_2$ was generated in the *flu* mutant background by transferring CL-grown plants to the dark for 8 h followed by re-illumination with varying lengths as indicated. It is important to note that EX1 (or modified EX1) was expressed under the control of the 35S promoter because EX1 driven by its native promoter demonstrates a too low abundance to perform PTM analysis. We have previously shown that there is no side effect by the overexpression of EX1-GFP on plant morphology and development since the plants behave like when using EX1-GFP driven by its native promoter[18].

**Vector construction and generation of transgenic plants**. The stop codon-less cDNA of EX1 and modified EX1 (EX1ΔSOS, EX1W643L, EX1W643A, EX1W565L, EX1F528C, and EX1G646D) were cloned into the pDONR221 Gateway vector (Invitrogen) through the Gateway BP reaction (Invitrogen) and subsequently recombined into the Gateway-compatible plant binary vector pGWB605/pGWB651 for C-terminal fusion with sGFP/G3GFP through the Gateway LR reaction (Invitrogen)[54]. An interlace-PCR-based approach was used to prepare amino acid substitution or domain deletion constructs for EX1. The generated vectors were transformed by electroporation into *Agrobacterium tumefaciens* strain GV3101. *Arabidopsis* transgenic plants were generated through *Agrobacterium*-mediated transformation by the floral dip method[55]. Transformants were selected on MS medium containing 20 mg/L glufosinate-ammonium (or BASTA; Sigma-Aldrich) until T3 generation to screen for homozygous transgenic plants.

**Co-immunoprecipitation (CoIP)**. Plants were homogenized in liquid nitrogen and the resulting powder was resuspended in CoIP buffer [containing 20 mM HEPES-KOH (pH 7.4), 2 mM EDTA, 2 mM EGTA, 25 mM NaF, 1 mM Na$_3$VO$_4$, 10% glycerol, 100 mM NaCl, 0.5% (w/v) Trition X-100, and SIGMAFAST$^{TM}$ Protease Inhibitor (1 tablet/100 mL)][18]. The suspension was kept on ice for 10 min before centrifugation at maximum speed (21,000 × *g*) in an Eppendorf table centrifuge for 30 min at 4 °C. The supernatant was filtered through a 0.22 μm Millipore Express PES membrane and proteins were quantified using Pierce$^{TM}$ BCA protein assay kit (Thermo Fisher Scientific). Small aliquots were taken as input samples, whereas the remaining parts were used for CoIP. GFP-conjugated beads (GFP-Trap-MA; ChromoTek) were added to equal amounts of the CoIP samples. The samples were first agitated at room temperature for 1.5–2 h, then the beads were washed four times for 5 min with wash buffer [containing 50 mM Tris-HCl (pH 7.5), 100 mM NaCl, 10% glycerol, 0.05% Triton X-100]. Finally, the washed beads were resuspended in 100 μl 1x Laemmli sodium dodecyl sulfate (SDS) sample buffer[56] and incubated for 20 min at 70 °C.

**Mass spectrometry, protein identification, and PTM analysis**. After CoIP enrichment, proteins were eluted from the beads using 6 M guanidine hydrochloride buffer (guanidine hydrochloride dissolved in 100 mM Tris pH 8.5) and denatured using 10 mM DTT at 56 °C for 30 min followed by alkylation in 50 mM iodoacetamide (IAA) at room temperature for 40 min in dark. The proteins were then desalted using a Nanosep membrane (Pall Corporation, MWCO 10 K) in 200 μL of 100 mM NH$_4$HCO$_3$ buffer. Desalted proteins were incubated in digestion buffer (40 ng/μL trypsin in 100 mM NH$_4$HCO$_3$, corresponding to the enzyme-to-protein ratio of 1:50) at 37 °C for 20 h. The digested peptides were dried and resuspended in 0.1% (v/v) formic acid (FA) solution. Digested peptides were separated using nanoAcquity Ultra Performance LC (Waters, Milford, MA, USA) and analyzed by using Q Exactive Mass Spectrometer (Thermo Fisher Scientific, San Jose, CA, USA). Briefly, digested peptides were separated by nanoAcquity Ultra Performance LC equipped with a 20 mm trap column (C18 5 μm resin, 180 μm I. D., Waters) and a 250 mm analytical column (C18 1.7 μm resin, 75 μm I.D., Waters). The separated peptides were analyzed by Q Exactive Mass Spectrometer. A full MS survey scan was carried out at a resolution of 70,000 at 400 *m/z* over the *m/z* range of 300–1800, with an automatic gain controls (AGC) target of 3E6 and a maximum ion injection time (IT) of 30 ms. The top 20 multiply-charged parent ions were selected by data-dependent MS/MS mode and fragmented by higher-energy collision dissociation (HCD) with a normalized collision energy of 27% at the *m/z* scan range of 200–2000. MS/MS detection was carried out at a resolution of 17,500 with the AGC target value of 5E5 and the maximum IT of 120 ms. Dynamic exclusion was enabled for 30 s.

The mass spectra were submitted to the Mascot Server (version 2.5.1, Matrix Science, London, UK) for peptide identification and scanned against the *Arabidopsis* protein sequences downloaded from TAIR website (http://www.arabidopsis.org/). Peptide mass tolerance was 20 ppm, fragment mass tolerance was 0.02 Da, and a maximum of two missed cleavages was allowed. Carbamidomethylation of Cys was set as a fixed modification while N-terminal acetylation, oxidation of Met, Trp, Tyr, His, and Cys, as well as phosphorylation of Ser, Trp, and Tyr were defined as variable modifications. The significance threshold for search results was set at a *P*-value of 0.05 and Ions score cutoff of 15.

For quantification, raw MS data files were processed and analyzed using MaxQuant software (version 1.5.8.3) with label-free quantitation (LFQ) algorithm[57,58]. Parent ion and MS2 spectra were searched against the FASTA database (http://www.arabidopsis.org/). The precursor ion tolerance was set at 7 ppm with an allowed fragment mass deviation of 20 ppm. Carbamidomethylation of Cys was set as a fixed modification while N-terminal acetylation, oxidation of Met, Trp, Tyr, and His as well as phosphorylation of Ser, Trp, and Tyr were defined as variable modifications. Peptides with a minimum of six amino acids and a maximum of two missed cleavages were allowed. False discovery rate (FDR) was set to 0.01 for both peptide and protein identification. The absolute intensity values were used to calculate the abundance of oxidized peptides.

**Isolation and fractionation of chloroplasts**. Intact chloroplasts were isolated from 3-week-old plants and separated into membrane and stroma fractions[18,59]. The rosette leaves of plants were homogenized in chloroplast isolation buffer [(50 mM Hepes-KOH, pH 8, 5 mM MgCl2, 5 mM EDTA pH8, 5 mM EGTA pH8, 10 mM NaHCO3, and 0.33 M D-sorbitol) supplemented with SIGMAFAST$^{TM}$ Protease Inhibitor (1 tablet/100 mL)] in a Waring blender. The homogenate was filtered through four-layers of Miracloth and centrifuged at 400 × *g* for 8 min at 4 °C. The pellets were suspended in isolation buffer and loaded onto a two-step Percoll gradient (40:80%) to separate intact and broken chloroplasts. Intact chloroplasts separated between the two Percoll steps were carefully collected and washed twice with HS buffer (50 mM Hepes-KOH, pH8 and 0.33 M D-sorbitol) by centrifugation at 2600 × *g* for 5 min at 4 °C. Chloroplasts corresponding to equal amounts of chlorophyll were resuspended in HM buffer (50 mM Hepes-KOH, pH 8 and 5 mM MgCl$_2$) and incubated for 10 min in ice, followed by a brief centrifugation at 2600 × *g* for 5 min at 4 °C to separate membrane and stroma fractions. To detect EX1-GFP proteins, chlorophyll was first removed using acetone[18] and enriched proteins were resuspended in 1x Laemmli SDS sample buffer and denatured for 10 min at 95 °C.

**Protein extraction and immunoblot analyses**. Total protein was extracted from 100 mg of plant tissues in CoIP buffer and quantified using the Pierce$^{TM}$ BCA protein assay kit (Thermo Fisher Scientific). Proteins were suspended in 1x Laemmli SDS sample buffer and incubated for 10 min at 95 °C. To detect modified EX1-GFP proteins (EX1W643A/L-GFP), at least 1200 μg proteins were enriched using the chloroform-methanol enrichment method and resuspended in a final volume of 100 μL 1x Laemmli SDS sample buffer and denatured for 10 min at 95 °C.

Equal amounts of the solubilized proteins from CoIP, fractionation, and total protein extractions were separated on 10% sodium dodecyl sulfate polyacrylamide gel electrophoresis gels and blotted onto polyvinylidene difluoride (PVDF) membranes (Bio-Rad). GFP fusion proteins were detected using a mouse anti-GFP monoclonal antibody (1:5,000; Roche, 11814460001). LhcB4, D1, D2, POR, FtsH2, RbcL, and UGP proteins were immunochemically detected using rabbit anti-LhcB4 (1:7,000; Agrisera, AS04 045), rabbit anti-D1 (1:10,000; Agrisera, AS05 084), rabbit anti-D2 (1:10,000; Agrisera, AS06 146), rabbit anti-POR (1:5,000; Agrisera, AS05 067), rabbit anti-FtsH (1:10,000; Agrisera, AS11 1789), rabbit anti-RbcL (1:10,000; Agrisera, AS03 037A), and rabbit anti-UGP (1:10,000; Agrisera, AS05 086) antibodies, respectively.

**RNA extraction and quantitative RT-PCR (qRT-PCR).** Total RNA was extracted using the Spectrum Plant Total RNA Kit (Sigma-Aldrich) and quantified spectrophotometrically at 260 nm using the NanoDrop 2000 (Thermo Fisher Scientific). One μg of RNA was treated with RQ1 RNase-Free DNase (Promega) and reverse-transcribed using Improm II reverse transcriptase (Promega) and oligo $(dT)_{15}$ primer (Promega) according to the manufacturer's recommendations. The qRT-PCR was performed with iTaq Universal SYBR Green PCR master mix (Bio-Rad) on a QuantStudio™ 6 Flex Real-Time PCR System (Applied Biosystems). Relative transcript levels were calculated using the comparative delta-Ct method and normalized to the transcript levels of ACTIN2 (At3g18780). The primers used in this study are listed in Supplementary Table 3.

**Confocal laser-scanning microscopy (CLSM).** For determination of the subcellular localization, healthy leaves of 3-week-old Nicotiana benthamiana Domin were infiltrated with Agrobacterium harboring native or modified EX1 proteins fused with GFP and observed under SP8 Confocal laser-scanning microscope (Leica Microsystems) after 48 h. Images were acquired and processed using Leica LAS AF Lite software version 2.6.3 (Leica Microsystems).

**Imaging protochlorophyllide and singlet oxygen.** Protochlorophyllide (Pchlide) accumulation in darkness was done according to Meskauskiene et al.[10]. Etiolated cotyledons of 5-day-old seedlings were exposed to blue light and the emitted fluorescence was recorded using the Leica M205 FA fluorescent microscope (Leica Microsystems). To detect $^1O_2$, 5-day-old seedlings grown under CL conditions were subjected to dark-to-light shift (8 h dark and 30 min light) and subsequently immersed in a solution of 260 μM Singlet Oxygen Sensor Green (SOSG; Thermo Fisher Scientific, Molecular Probes) in 50 mM phosphate buffer (pH 7.4). Leaves were vacuum infiltrated for 5 min and then incubated for 2 h in dark followed by imaging using the GFP filter of the Leica M205 FA fluorescent microscope (Leica Microsystems). $^1O_2$-activated SOSG was visualized with excitation at 488 nm and emission at 530 nm. At least ten cotyledons from each genotype were monitored and representative images were shown. All the images were acquired and processed using Leica LAS software version 4.2.0 (Leica Microsystems).

## Data availability

The authors declare that all data supporting the findings of this study are available within the manuscript and its supplementary files. The source data for all the plots and uncropped pictures of all the blots presented in the main and Supplementary Figures are provided as a Source Data file. The datasets generated and analyzed during the current study are available from the corresponding author on reasonable request.

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

## Acknowledgements

We thank the Core Facility of Proteomics and Dr. Rongxia Li, Shanghai Center for Plant Stress Biology (PSC) for carrying out mass-spectrometric analysis. This research was supported by the Strategic Priority Research Program from the Chinese Academy of Sciences (Grant No. XDB27040102), the 100-Talent Program of the Chinese Academy of Sciences and the National Natural Science Foundation of China (NSFC) (Grant No. 31871397) to C.K. The supports from Research Fund for International Young Scientists Program of NSFC (Grant No. 31850410478) and President's International Fellowship Initiative (PIFI) postdoctoral fellowship from the Chinese Academy of Sciences (No. 2019PB0066) to V.D. are also acknowledged.

## Author contributions

V.D. and C.K. designed the research; V.D., M.L., S.S. and M.P.L. conducted the experiments; V.D., S.S. and C.K. analyzed the data; V.D. and C.K. wrote the manuscript. All authors reviewed the manuscript.

## Additional information

**Competing interests:** The authors declare no competing interests.

