## [Peer Review File · Nature Communications]

Reviewers' comments:

Reviewer #1 (Remarks to the Author):

This is a very interesting work by Dogra et al that explores the mode of action of Executer 1 proteins. Executer 1 participates in singlet oxygen-mediated retrograde signaling in chloroplasts, but its molecular properties are currently unknown. From the experiments presented in this work, authors concluded that dark/light cycles promote Executer1 degradation as a consequence of the oxidation of a tryptophan amino acids by singlet oxygen. It seems that the FtsH2 protease is involved in the proteolysis of Executer 1. Authors identified the oxidized tryptophan by mass spectrometry. Amino acid sequence analysis of the protein family reveals that the tryptophan forms part of a protein region that authors classify as the singlet oxygen sensor (SOS) domain. Deletion of the SOS domain protects the truncated protein from degradation and results in its accumulation into the stroma. Mutation of the tryptophan or deletion of the SOS domain affect Executer 1 properties and revoke the Executer1-mediated singlet oxygen signaling. In conclusion, the Executer 1 protein senses the singlet oxygen by oxidation of a tryptophan with subsequent proteolytic degradation by FtsH2.

This is a fascinating proposal, but I would like to include some points for discussion.

The manuscript is based on the suggestion of a functional association between tryptophan-oxidized Ex1 and the FtsH2 protease. However, authors do not present any direct evidence for this suggestion that seems to be in many occasions based on some speculation. Have the authors consider effects on Executer 1 protein turnover during dark/light cycles or protein aggregation due to tryptophan oxidation?

Due to the lack of a useful antibody for Executer 1, the protein content has been analyzed in plants transformed with Executer 1 attached to a GFP protein at its C-terminus. GFP is a medium size protein that has been fused to Executer 1 by a flexible linker, that might have a destabilizing effect on stability. Moreover, authors have selected a strong promoter for expression that might not reflect the physiological protein content in plants. Plants have been grown under continuous light conditions. Authors should comment on the selected methodological approximations.

In the manuscript, conclusions are based on the assumption that the tryptophan mutations avoid oxidation, but authors should explore other explanations, such as that mutations of tryptophan-643 to alanine or leucine result in aberrant proteins.

The proteins found to be associated with Executer 1 (Supplementary Table 1) needs further biochemical confirmation. For example, one of detected proteins is Tic110, a membrane protein at the inner envelope of chloroplast. Is this a false positive, or are authors aware of a functional relation with Executer 1? This part of the work is in a close connection with a previous published paper of the same authors. I would suggest a comment on the differences between the two papers.

The sentence in line 251 should be improved. The main role of tryptophan amino acids in proteins is not to mediate interactions with membranes.

Fig. 2a: The Executer1 protein content seems to be very similar in the input sample for DD and CL conditions. Is this observation in accordance with the conclusions of this work?

Reviewer #2 (Remarks to the Author):

In this manuscript the authors present compelling evidence that a specific Trp in the protein EX1 is oxidized in a light- and singlet oxygen-dependent manner and that this leads to the degradation of EX1 followed by an activation of singlet oxygen-dependent genes. They further show that substitution of this Trp with singlet oxygen-resistant amino acids abolishes this chain of events.

The experiments appear to have been carried out very competently, the results are new and interesting and the presentation is, on the whole, quite good although the English requires a bit of attention. I think that the manuscript can be accepted after a relatively light revision.

One comment for the Discussion:

Tryptophan oxidation has been described relatively rarely in any system and I think that it would make sense to mention two studies on tryptophan oxidation in mitochondria, another major cellular site of detection of environmental signals with ROS production and retrograde signaling to the nucleus:

Moller, I.M., Kristensen, B.K. (2006) *Free Radical Biology & Medicine* 40, 430 – 435

Taylor, S. W.; Fahy, E.; Murray, J.; Capaldi, R. A.; Ghosh, S. S. (2003) *J. Biol. Chem.* 278,19587–19590

I have entered a number of comments, questions and corrections, most of them of an editorial nature, directly in a copy of the manuscript. This copy has been scanned and the pdf-file is attached for use by the editor and the authors.

Reviewer #3 (Remarks to the Author):

In this manuscript, the authors report that a specific tryptophan residues (W643) of the chloroplast EXECUTER1 (X1) protein, previously shown to be involved in transmitting singlet oxygen signals, undergoes oxidation in a light-dependent manner that is probably mediated by singlet oxygen. The manuscript then describes attempts to establish the functional importance of this modification for responses to singlet oxygen. It is shown that deletion of the EX1 DUF3506 domain (here renamed SOS domain), which includes the W643 residue, compromises the ability of EX1 to complement the ex1 loss-of-function mutation. This is consistent with a role for W643 in the EX1 signaling function. However, attempts to provide more direct evidence for the importance of W643 oxidation by using site-directed mutagenesis lead to equivocal results. The engineered mutant proteins appear to be unstable in planta, making it unclear whether their failure to complement the ex1 mutation is related to the loss of W643 and its oxidation, or rather to effects of the mutations on protein stability that may not be related to the protein's biological function in the plant.

Perhaps I have misunderstood parts of the hypothesis, and perhaps the manuscript could benefit from a cartoon to illustrate the proposed mechanism more clearly. A significant part of the text, and most notably the introduction, is taken up by an issue for which little or no relevant data are presented, ie, the possible importance of EX1 localization within the thylakoids. By contrast, exactly how the reported EX1 oxidation integrates with EX1 degradation in singlet oxygen signaling remains enigmatic. This issue is not elucidated clearly enough in the interpretation and the overarching advance remains uncertain without a more direct demonstration that the singlet oxygen-induced modification of W643 is an important process in EX1 function.

Specific points

(1) Perhaps the manuscript has omitted to explain significant details necessary to understand how EX1 is thought to work, but I am left with several questions. If singlet oxygen signaling proceeds via oxidant-induced degradation of EX1, why is signaling abrogated (rather than constitutive) in ex1 mutants, in which EX1 function is lost either because of a modified or absent protein, as originally described by Wagner et al. (2004)? If EX1 degradation is a necessary part of the EX1 signaling function why does introducing mutated proteins that accumulate to very low levels fail to complement the ex1 mutation (Fig. 4)?

(2) L292. "Three missense mutations in EX1 were found to repress 102 signaling. Among them, two were located in the SOS domain resulting in the substitution of Phe528Cys and Gly646Asp, respectively. Given the proximity to Trp643, Gly646Asp may induce a structural change and

possibly may affect interaction between Trp643 and 1O₂." Since the substitution of W643 with A or L has not, to my mind, generated conclusive results, useful evidence could come from testing this hypothesis experimentally.

(3) L323. "Trp643 oxidation and the abrogation of 1O₂ signaling by substituting it with a 1O₂-insensitive amino acid or SOS domain deletion resemble the mode of action of the chloroplastic hydrogen peroxide (H₂O₂) sensor 3'-phosphoadenosine 5'-phosphate (PAP) phosphatase SAL1." The resemblance is not clear to me. First, the oxidant-sensitive amino acids are quite different. Second, unlike SAL1, EX1 does not seem to have a clearly identified substrate.

(4) Statistics are largely missing from the manuscript. Some of the claims should be backed up by statistical analysis of replicates; eg, effect of flu mutation on EX1 protein abundance (Fig 1c), change in oxidized W643 signal (Fig. 2c), protein abundance (Fig. 4g).

Reviewer #1 (Remarks to the Author):

This is a very interesting work by Dogra et al. that explores the mode of action of Executer 1 proteins. Executer 1 participates in singlet oxygen-mediated retrograde signaling in chloroplasts, but its molecular properties are currently unknown. From the experiments presented in this work, the authors concluded that dark/light cycles promote Executer 1 degradation as a consequence of the oxidation of a tryptophan amino acid by singlet oxygen. It seems that the FtsH2 protease is involved in the proteolysis of Executer 1. The authors identified the oxidized tryptophan by mass spectrometry. Amino acid sequence analysis of the protein family reveals that the tryptophan forms part of a protein region that the authors classify as the singlet oxygen sensor (SOS) domain. Deletion of the SOS domain protects the truncated protein from degradation and results in its accumulation into the stroma. Mutation of the tryptophan or deletion of the SOS domain affect Executer 1 properties and revoke the Executer1-mediated singlet oxygen signaling. In conclusion, the Executer 1 protein senses the singlet oxygen by oxidation of a tryptophan with subsequent proteolytic degradation by FtsH2. This is a fascinating proposal, but I would like to include some points for discussion.

The manuscript is based on the suggestion of a functional association between tryptophan-oxidized EX1 and the FtsH2 protease. However, authors do not present any direct evidence for this suggestion that seems to be in many occasions based on some speculation. Have the authors considered effects on Executer 1 protein turnover during dark/light cycles or protein aggregation due to tryptophan oxidation?

Response: We thank the reviewer for this suggestion. Providing direct evidence for the functional association between the oxidized EX1 and FtsH protease is practically very difficult. However, the facts that EX1 undergoes degradation in response to $^1\text{O}_2$ through FtsH protease and that both EX1 and FtsH protease are localized and associated in the grana margin (Fig. 1a; Wang et al., 2016) indicate the functional association of EX1 and FtsH protease in initiating $^1\text{O}_2$ signaling. All data presented here suggest that thylakoid membrane-bound FtsH protease is responsible for the EX1 degradation upon its oxidation (or aggregation). We now added the result regarding the FtsH-dependent EX1 degradation (Supplementary Fig. 1; lines 108-110).

Due to the lack of a useful antibody for Executer 1, the protein content has been analyzed in plants transformed with Executer 1 attached to a GFP protein at its C-terminus. GFP is a medium size protein that has been fused to Executer 1 by a flexible linker, that might have a destabilizing effect on stability. Moreover, authors have selected a strong promoter for expression that might not reflect the physiological protein content in plants. Plants have been grown under continuous light conditions. Authors should comment on the selected methodological approximations.

Response: Due to the absence of an EX1-specific antibody, we used GFP-tagged EX1. In our previous study, we have shown that EX1-GFP expressed under the control of the 35S promoter complements *ex1 flu* (Wang et al., 2016). In addition, we also demonstrated the rapid degradation of Myc-tagged EX1 in response to $^1\text{O}_2$ (Wang et al., 2016). Furthermore, our results indicated a specific $^1\text{O}_2$ -dependent EX1 degradation and the overexpression of

EX1 has no side effect on plant morphology and development (Wang et al., 2016). Since EX1 driven by the native promoter is hard to detect by MS analysis, we had to use the overexpression line for PTM analysis (lines 381-385). We used continuous light condition since the *flu* mutant generates lethal levels of $^1\text{O}_2$ upon a dark-to-light transition.

In the manuscript, conclusions are based on the assumption that the tryptophan mutations avoid oxidation, but authors should explore other explanations, such as that mutations of tryptophan-643 to alanine or leucine result in aberrant proteins.

Response: We thank the reviewer for this suggestion. Yes, it is possible that these modified EX1 proteins might result in aberrant proteins, which lead to their instability. This concern can be resolved once we reveal the EX1 structure, an experimental process we now initiated. Future elucidation of the EX1 protein structure may also shed light on the effects of the substitutions introduced in this study (lines 328-334). In this revised manuscript, we alternatively corroborate the importance of W643 oxidation by examining the presence or absence of W643 oxidation in the *ex1 flu* transgenic lines expressing EX1F528C and EX1G646D (Fig. 5; Supplementary Figs. 5 and 6; Supplementary Table S3; lines 263-277, 320-328). These substitutions impair $^1\text{O}_2$ signaling (lines 320-334) (Wagner et al., 2004).

The proteins found to be associated with Executer 1 (Supplementary Table 1) need further biochemical confirmation. For example, one of the detected proteins is Tic110, a membrane protein at the inner envelope of chloroplasts. Is this a false positive, or are authors aware of a functional relation with Executer 1? This part of the work is in a close connection with a previous published paper of the same authors. I would suggest a comment on the differences between these two papers.

Response: IP-MS analysis was performed using the same method as described earlier (Wang et al., 2016). Most of the IP-enriched proteins we also identified in our previous study with a minor difference, which could be attributed to the age of the tissue used. Among those identified proteins, the PSII RC proteins D1 and D2, chlorophyll biosynthetic enzymes PORB/C, and the FtsH protease were validated by IP/western analysis (Fig. 2a). Besides these proteins, some other proteins were also enriched including Tic110 (Supplementary Table 1). We routinely perform IP-MS analyses using chloroplast proteins as baits. As a result, we noticed that Tic110 tend to appear in almost every experiment, possibly due to its abundance and stickiness as an intrinsic membrane protein. Hence, it is possible that Tic110 might be a false positive.

The sentence in line 251 should be improved. The main role of tryptophan amino acids in proteins is not to mediate interactions with membranes.

Response: The statement has been omitted and the overall sentence is rephrased. (lines 280-281).

Comment: Fig. 2a: The Executer1 protein content seems to be very similar in the input sample for DD and CL conditions. Is this observation in accordance with the conclusions of this work?

Response: We used equal amounts of proteins from DD (etiolated seedlings) and CL (cotyledons) samples as quantified by BCA, however, the biasness due to the presence of pigments in the protein samples could not be ruled out. Nevertheless, this observation has no direct implications on the conclusions of the study.

Reviewer #2 (Remarks to the Author):

In this manuscript the authors present compelling evidence that a specific Trp in the protein EX1 is oxidized in a light- and singlet oxygen-dependent manner and that this leads to the degradation of EX1 followed by an activation of singlet oxygen-dependent genes. They further show that substitution of this Trp with singlet oxygen-resistant amino acids abolishes this chain of events.

The experiments appear to have been carried out very competently, the results are new and interesting and the presentation is, on the whole, quite good although the English requires a bit of attention. I think that the manuscript can be accepted after a relatively light revision. I have entered a number of comments, questions and corrections, most of them of an editorial nature, directly in a copy of the manuscript. This copy has been scanned and the pdf-file is attached for use by the editor and the authors.

Response: We thank the reviewer for the suggestions. We modified the manuscript accordingly (highlighted in bold).

Comment: One comment for the Discussion: Tryptophan oxidation has been described relatively rarely in any system and I think that it would make sense to mention two studies on tryptophan oxidation in mitochondria, another major cellular site of detection of environmental signals with ROS production and retrograde signaling to the nucleus: Moller, I.M., Kristensen, B.K. (2006) Free Radical Biology & Medicine 40, 430 – 435 Taylor, S. W.; Fahy, E.; Murray, J.; Capaldi, R. A.; Ghosh, S. S. (2003) J. Biol. Chem. 278,19587–19590.

Response: Thank you for pointing us to these interesting publications. These references have been added to our manuscript (also see lines 173-174).

In Fig. 1a and b, band intensities of LhcB4 and UGP could also be plotted here or in supplementary data.

Response: Band intensities have been added in Fig. 1c

In Fig. 3, DUF3506 (SOS domain) has 131 residues, but why does the multiple alignment (Fig. 3b) show 134 residues?

Response: In Arabidopsis, the SOS domain of EX1 comprises 131 residues. For understanding the conservation of the SOS domain across the plant species, we used SOS domain sequences from in total 27 plant species among which some have gaps in between. In multiple alignments, the gaps are also counted which increase the count to 134 residues.

In Fig. 5e, an increase? Reproducible?

Response: Yes. We observed a slight increase which was reproducible since we observed this increase in all our experiments.

Reviewer #3 (Remarks to the Author):

In this manuscript, the authors report that a specific tryptophan residue (W643) of the chloroplast EXECUTER1 (EX1) protein, previously shown to be involved in transmitting singlet oxygen signals, undergoes oxidation in a light-dependent manner that is probably mediated by singlet oxygen. The manuscript then describes attempts to establish the functional importance of this modification for responses to singlet oxygen. It is shown that deletion of the EX1 DUF3506 domain (here renamed SOS domain), which includes the W643 residue, compromises the ability of EX1 to complement the *ex1* loss-of-function mutation. This is consistent with a role for W643 in the EX1 signaling function. However, attempts to provide more direct evidence for the importance of W643 oxidation by using site-directed mutagenesis lead to equivocal results. The engineered mutant proteins appear to be unstable *in planta*, making it unclear whether their failure to complement the *ex1* mutation is related to the loss of W643 and its oxidation, or rather to effects of the mutations on protein stability that may not be related to the protein's biological function in the plant.

Perhaps I have misunderstood parts of the hypothesis, and perhaps the manuscript could benefit from a cartoon to illustrate the proposed mechanism more clearly.

Response: Thank you for pointing out that some misunderstanding might exist which could be abrogated by adding an illustrated proposed mechanism. We now added the working model illustrating the possible effect of W643 oxidations in EX1 towards $^1\text{O}_2$ signaling in Fig. 6.

Comment: A significant part of the text, and most notably the introduction, is taken up by an issue for which little or no relevant data are presented, ie, the possible importance of EX1 localization within the thylakoids. By contrast, exactly how the reported EX1 oxidation integrates with EX1 degradation in singlet oxygen signaling remains enigmatic. This issue is not elucidated clearly enough in the interpretation and the overarching advance remains uncertain without a more direct demonstration that the singlet oxygen-induced modification of W643 is an important process in EX1 function.

Response: We thank the reviewer for this suggestion. We previously reported the importance of EX1 degradation in initiating $^1\text{O}_2$ signaling in the *var2* (*ftsh2*) mutant background (Wang et al., PNAS 2016; Dogra et al., Frontiers in Plant Science; Supplementary Fig. 1 in this study). We further corroborate the importance of W643 oxidation using plant lines with Trp643 substitution, SOS domain-deletion, and by examining the presence or absence of W643 oxidation in the *ex1 flu* transgenic lines expressing EX1F528C and EX1G646D (Fig. 5; Supplementary Fig. 5 and 6; Supplementary Table S3; lines 263-277, 320-328).

Specific points

(1) Perhaps the manuscript has omitted to explain significant details necessary to understand how EX1 is thought to work, but I am left with several questions. If singlet oxygen signaling proceeds via an oxidant-induced degradation of EX1, why is signaling abrogated (rather than constitutive) in *ex1* mutants, in which EX1 function is lost either because of a modified or absent protein, as originally described by Wagner et al. (2004)? If EX1 degradation is a necessary part of the EX1 signaling function why does introducing mutated proteins that accumulate to very low levels fail to complement the *ex1* mutation (Fig. 4)?

Response: The EX1 oxidation as well as the EX1-associated protein complex seem to be required to initiate $^1\text{O}_2$ signaling, meaning that such a surveillance complex must be present to respond appropriately to $^1\text{O}_2$. Owing to its constant degradation, it is very likely that EX1W643L/A-GFP proteins cannot form such a complex to monitor $^1\text{O}_2$, which is the case in *ex1 flu*. This could be one reason why EX1W643L/A-GFP is unable to mediate $^1\text{O}_2$ signaling (lines 228-238; Fig. 6).

(2) L292. “Three missense mutations in EX1 were found to repress $^1\text{O}_2$ signaling. Among them, two were located in the SOS domain resulting in the substitution of Phe528Cys and Gly646Asp, respectively. Given the proximity to Trp643, Gly646Asp may induce a structural change and possibly may affect interaction between Trp643 and $^1\text{O}_2$.” Since the substitution of W643 with A or L has not, to my mind, generated conclusive results, useful evidence could come from testing this hypothesis experimentally.

Response: Thanks for this valuable suggestion. Fortunately, one of our postdocs was preparing the corresponding transgenic lines EX1F528C-GFP and EX1G646D-GFP in *ex1 flu*, which became available recently. Therefore, we tested this hypothesis with no considerable delay and the results further corroborated that W643 oxidation is required to initiate the $^1\text{O}_2$ signaling (Fig. 5; Supplementary Fig. 5 and 6; Supplementary Table S3; lines 263-277, 320-328).

(3) L323. “Trp643 oxidation and the abrogation of $^1\text{O}_2$ signaling by substituting it with a $^1\text{O}_2$ -insensitive amino acid or SOS domain deletion resemble the mode of action of the chloroplastic hydrogen peroxide (H_2O_2) sensor 3'-phosphoadenosine 5'-phosphate (PAP) phosphatase SAL1.” The resemblance is not clear to me. First, the oxidant-sensitive amino acids are quite different. Second, unlike SAL1, EX1 does not seem to have a clearly identified substrate.

Response: The sentences has been corrected and rephrased. (see lines 35-362).

(4) Statistics are largely missing from the manuscript. Some of the claims should be backed up by statistical analysis of replicates; eg, effect of *flu* mutation on EX1 protein abundance (Fig. 1c), change in oxidized W643 signal (Fig. 2c), protein abundance (Fig. 4g).

Response: We now added the statistical analysis.

REVIEWERS' COMMENTS:

Reviewer #3 (Remarks to the Author):

In their revised manuscript, the authors have addressed my comments clearly, and the figures and text have been improved. The notion of a "surveillance" process, whereby the process of degradation itself is somehow monitored, is very interesting and consistent with the data, even if further work will be required to establish whether this is indeed the case and, if so, exactly how it works.

REVIEWERS' COMMENTS:

Reviewer #3 (Remarks to the Author):

In their revised manuscript, the authors have addressed my comments clearly, and the figures and text have been improved. The notion of a “surveillance” process, whereby the process of degradation itself is somehow monitored, is very interesting and consistent with the data, even if further work will be required to establish whether this is indeed the case and, if so, exactly how it works.

Response: Thanks for your encouraging comment. As you consider, we will continue to work on this hypothesis to refine whether Trp643 oxidation is a part of the surveillance system.